# Conservative Bladder Management and Medical Treatment in Chronic Spinal Cord Injury Patients

**DOI:** 10.3390/jcm12052021

**Published:** 2023-03-03

**Authors:** Hueih Ling Ong, I-Ni Chiang, Lin-Nei Hsu, Cheih-Wen Chin, I-Hung Shao, Mei-Yu Jang, Yung-Shun Juan, Chung-Cheng Wang, Hann-Chorng Kuo

**Affiliations:** 1Department of Urology, Dalin Tzu Chi Hospital, Buddhist Tzu Chi Medical Foundation, Chia-Yi 622, Taiwan; 2Department of Urology, National Taiwan University Hospital, College of Medicine, National Taiwan University, Taipei 110, Taiwan; 3Department of Urology, An Nan Hospital, China Medical University, Tainan City 833, Taiwan; 4Feng Shan Lee Chia Wen Urologic Clinic, Kaohsiung 800, Taiwan; 5Division of Urology, Department of Surgery, Chang Gung Memorial Hospital, Linkou Branch, Taoyuan 333, Taiwan; 6Department of Urology, Kaohsiung Municipal Siaogang Hospital, Kaohsiung 812, Taiwan; 7Department of Urology, College of Medicine, Kaohsiung Medical University, Kaohsiung 807, Taiwan; 8Department of Urology, Kaohsiung Medical University Hospital, Kaohsiung 813031, Taiwan; 9Graduate Institute of Clinical Medicine, College of Medicine, Kaohsiung Medical University, Kaohsiung 807, Taiwan; 10Department of Urology, En Chu Kong Hospital, New Taipei City 237, Taiwan; 11Department of Biomedical Engineering, Chung Yuan Christian University, Chungli 320, Taiwan; 12Department of Urology, Hualien Tzu Chi Hospital, Buddhist Tzu Chi Medical Foundation, Buddhist Tzu Chi University, Hualien 970, Taiwan

**Keywords:** spinal cord injury, neurogenic bladder, neuromodulation, quality of life, treatment, botulinum injection, catheterization

## Abstract

To review the available data on non-surgical management for neurogenic lower urinary tract dysfunction (NLUTD) in patients with chronic spinal cord injury (SCI) and provide the most updated knowledge for readers. We categorized the bladder management approaches into storage and voiding dysfunction separately; both are minimally invasive, safe, and efficacious procedures. The main goals for NLUTD management are to achieve urinary continence; improve quality of life; prevent urinary tract infections and, last but not least, preserve upper urinary tract function. Annual renal sonography workups and regular video urodynamics examinations are crucial for early detection and further urological management. Despite the extensive data on NLUTD, there are still relatively few novel publications and there is a lack of high-quality evidence. There is a paucity of new minimally invasive and prolonged efficacy treatments for NLUTD, and a partnership between urologists, nephrologists and physiatrists is required to promote and ensure the health of SCI patients in the future.

## 1. Introduction

Lower urinary tract (LUT) dysfunction is a common sequela to spinal cord injury (SCI) patients. It might be presented with a wide range of LUT symptoms depending on the level of SCI, and the symptoms vary as time goes by [1]. Its heterogenous presentation and the individualized treatment needed for each SCI patient becomes extremely arduous as the standard treatment modalities often have limitations. Moreover, pelvic organ dysfunction encompasses LUT, sexual and bowel dysfunction and their complex interrelationship is crucial for a comprehensive approach for management in SCI patients [2]. LUT dysfunction in SCI patients can be life threatening or a sequela to upper urinary tract complications in the chronic phase. The negative effects of LUT dysfunction on patients’ health and quality of life are extremely severe and have been of great interest for researchers and clinicians for several decades [3]. Goals of management are none other than to achieve urinary continence, improve quality of life (QoL), prevent urinary tract infections (UTI), and preserve upper urinary tract function [4].

Lesions of the nervous systems, either peripheral or central and the level of lesion in the spinal cords can result in different patterns of neurogenic LUT dysfunction (NLUTD). The lesions of suprapontine or spinal pathways regulating LUT functions affect the storage phase, causing reduced bladder capacity and detrusor overactivity (DO), leading to involuntary contraction of the detrusor muscle during the storage phase [5]. Patients might present with urinary urgency, frequency, nocturia, and urge urinary incontinence. Injury to infrapontine-suprasacral spinal pathways can result in loss of coordination between detrusor contraction and urethral sphincter relaxation during the voiding phase, which is known as detrusor sphincter dyssynergia (DSD) [6]. Patients might also present with voiding difficulty and incomplete bladder emptying, which can result from high intravesical pressure and DSD. DO in combination with DSD can cause high intravesical pressure during the voiding phase, which then leads to morphological changes of the bladder wall [7]. Long-term morphological changes of the bladder wall due to neuropathy and outlet obstruction can reduce bladder compliance and bladder capacity, and consequently increase the risk of upper urinary tract complications such as vesico-uretero-renal reflux, hydronephrosis, upper urinary tract UTI, renal impairment and eventually end-stage renal disease [8]. Lesions of the sacral cord or infrasacral pathways can result in voiding dysfunction, detrusor underactivity (DU) or acontractile detrusor associated with a loose or non-relaxing sphincter [9].

NLUTD in SCI patients is mainly due to either storage dysfunction and/or voiding dysfunction. In this review we aimed to underpin bladder management especially in the aspects of conservative and medical treatment for chronic SCI patients. In order to construct a review of the conservative management and medical treatment of chronic SCI patients, we searched PUBMED articles published from January 2002 to August, 2022, with the search terms “antimuscarinics”, “botulinum toxin”, “catheterization”, “desmopressin”, “neurogenic bladder”, “neurogenic lower urinary tract dysfunction”, “tibial nerve stimulation”, “β3-Adrenoceptor agonists”, “Credé maneuver”, “Valsava maneuver”, “α-adrenoceptor blockers”, and “urodynamics”. There were no language restrictions. We also added some important references published before 2022, and the final reference list was generated based on relevance to the topics.

## 2. Goal of Bladder Management

To achieve urinary continence, improved QoL, and preserve the upper urinary tract function, the management of NLUTD should address both storage and voiding dysfunction. Urinary incontinence is a prevalent problem in patients with chronic SCI, especially in women, affecting about half of the patients and is associated with impaired mobility and reduced quality of life [10]. Patients with storage dysfunction are presented with overactive bladder symptoms such as urgency, frequency, nocturia, with or without urge urinary incontinence. Moreover, some patients present with stress urinary incontinence due to urethral sphincter or bladder neck incompetence. However, untreated voiding dysfunction eventually has an impact on renal function [8].

## 3. Management of Storage Dysfunction

### 3.1. Physical Therapy

The first-line treatment for storage dysfunction is behavioral intervention, to correct unhelpful habits such as periodic massive drinking, frequent voiding and caffeine consumption. Behavioral therapies require support from caregivers and healthcare professionals for guidance and consultation [11]. Timed voiding based on timed water intake of a specific amount reduces involuntary bladder emptying. Weight reduction reduces the intra-abdominal pressure that causes incontinence in patients with urethral or bladder neck incompetence [12].

Pelvic floor muscle training (PFMT) is a well-established conservative treatment for urinary continence, with a five times higher symptomatic cure rate compared to the control group [13]. PMFT increases urethral pressure, supports the bladder neck and improves coordination between pelvic floor muscle and transversus abdominis, which results in improved bladder function [14,15].

### 3.2. Antimuscarinic Drugs

The mainstay of pharmacotherapy for storage problems in NLUTD patients has, for decades, been anticholinergic medications that facilitate storage of urine. Anticholinergics have been recommended by the European Association of Urology [16] and a UK consensus group [17] for this purpose. Antimuscarinic agents are subtype M2 and M3 muscarinic receptor antagonists. They are used to stabilize the detrusor, reducing its overactivity, and making it moderately refractory to parasympathetic stimulation. They improves bladder compliance, and are associated with reduction of symptoms, prevention of upper urinary tract damage and improvement of QoL in neurogenic DO [18]. Oxybutynin chloride, tolterodine, darifenacin, trospium, and propiverine are established effective medical treatments for neurogenic detrusor overactivity (NDO) [19]. Among anticholinergics, trospium chloride, a mixed antimuscarinic agent and a smooth muscle relaxant, has been shown to improve cystometric capacity and reduce incontinence episodes, with a lower risk of cognitive impairment in the elderly [20]. However, bothersome side effects from these medications could be a concern [21,22]. In one study, only 18% of patients received pharmacotherapy without combining it with urinary catheterizations [23]; this may imply the limited role of pharmacotherapy as a single and main therapy.

### 3.3. Desmopressin

Desmopressin is the synthetic analogue of the human hormone vasopressin, an antidiuretic hormone that acts on V2 receptors in the distal collecting tubules, which promote free water passive reabsorption from nephrons back into the systemic circulation. It is mostly frequently used for central diabetes insipidus, bleeding disorder and primary nocturnal enuresis [24]. SCI patients often have a high urine output at night, particularly those with higher spinal cord level lesions [25]. Nocturnal polyuria is multifactorial with fluid retention during the day secondary to autonomic dysfunction, lack of ambulation and arginine vasopressin production disorder [26]. Two small studies have assessed desmopressin in patients with SCI with nocturnal polyuria; 11/15 patients eliminated the need of clean intermittent catheterization (CIC) at night and the remaining 4 patients had CIC only once at night. No patient suffered from significant side effects of hyponatremia or fluid retention [26,27]. A systematic review suggested that desmopressin is useful for treating nocturia in patients with multiple sclerosis, the rate of hyponatremia ranged from 0% to 23.5% [28]. All patients need a baseline serum creatinine and sodium level measurement before starting therapy. Serum sodium measurement should be repeated within a week as hyponatremia is the most feared side effect that usually presents with confusion, hallucination, or seizure [29].

### 3.4. β3-Adrenoceptor Agonists

The human bladder contains 3 beta adrenergic receptors (β1, β2 and β3), 97% of the beta receptors being β3. Mirabegron is a β3 receptor agonist that relaxes the detrusor muscle, and it is an ideal target for the treatment of DO [30]. A systematic review suggested mirabegron as a second line treatment after antimuscarinics due to lack of: efficacy or adverse effects. There are reported improvements in bladder compliance and maximal cystometric capacity, and in IPSS subscores particularly in storage symptoms [31]. A meta-analysis reported no significant differences of drug-related adverse events between mirabegron and control groups in patients with NLUTD, such as arrythmia, hypertension, or post-voiding residual volume (PVR) [32].

### 3.5. Neuromodulation

Tibial nerve stimulation (TNS) was introduced by Stoller in the late 1990s [33], and was reported to be safe and effective for treating idiopathic overactive bladder in randomized controlled trials (RCTs) [34,35,36]. The treatment course consists of stimulating the nerve through a fine gauge stainless steel needle using a 34-gauge fixed frequency electrical signal, once weekly for 30 min, over a 12-week period [37]. Schneider et al. performed a systematic review of the literature which included 16 studies, 469 patients [38]. Five studies report on acute TNS (stimulation during urodynamic investigation) whereas 11 report on chronic TNS (weekly stimulation for 6–12 weeks before urodynamic investigation). In both acute and chronic TNS, studies report increases in mean maximum cystometric capacity, bladder volume at first DO and decreases in mean maximum detrusor pressure during the storage phase. In chronic TNS, the mean decrease in number of voids per 24h, in number of leakages per 24h, and in PVR ranged from 3 to 7, from 1 to 4, and from 15 to 55 mL, respectively. No TNS-related adverse events have been reported. However, the risk of bias and confounding was high in most studies.

A recently published study examined the use of percutaneous TNS (PTNS) in complete SCI patients [39]. This study randomized 100 individuals to either PTNS or solifenacin. At 2 weeks, there was no significant improvement in maximum functional capacity, leakage per day, and QoL between the groups. However, PTNS was noted to be more tolerable with lesser side effects. Early transcutaneous tibial nerve stimulation in rat shows a decrease in the number of non-voiding bladder contractions and maximum intravesical pressure during storage and an increase in voiding efficiency. However, the effects were not sustained after discontinuation of neurostimulation. Interestingly, neuromodulation has no safeguards in the development of DSD [40]. Although TNS appears to be a promising and novel treatment for neurogenic LUT dysfunction, there is a need for more reliable data to reach definitive conclusions.

Sacral neuromodulation (SNM) has been tried to treat NLUTD in patients with any kind of neurological lesions. Currently, most of the evidence is based on small sample sizes and different patient groups, including SCI, multiple sclerosis, cerebrovascular accident, and spina bifida [41]. The early overall results of SNM on NLUTD are promising in selected cases. However, in SCI patients, SNM has been shown ineffective in complete SCI, but was clinically beneficial in the spinal shock phase with early treatment by bilateral SNM [42].

### 3.6. Botulinum Toxin (Botox)

#### Detrusor Botox Injection Treatment

Botox, which was first isolated in 1897 by van Ermengem, is a potent neurotoxin produced by Clostridium botulinum [43]. It has a role in the neuromuscular junction that inhibits acetylcholine neurotransmitter release resulting in striated muscle relaxation [44]. However, increasing evidence shows that Botox might also inhibit the release of other neurotransmitters such as acetylcholine, adenosine triphosphate, and substance P. It may also down-regulate the expression of purinergic and capsaicin receptors on afferent neurons within the bladder [45]. Therefore, Botox has an important role in treating DO by both motor and sensory pathways. It has been developed as a second-line treatment option for patients with non-neurogenic DO with urinary incontinence, refractory to antimuscarinics or β3-adrenoceptor agonists [46]. Karsenty et al. performed a systematic review including 18 articles with a total of 698 patients, which evaluates the efficacy and safety of intradetrusor Botox injection in patients with wet NDO with or without CIC. Mostly with the amount of 300U Botox injection, a reported 40–80% of patients became completely dry with combined CIC; and mean maximum detrusor pressure was reduced to ≤40 cm H_2_O, QoL and no major adverse events were reported [47].

A 3-year extension study included multiple sclerosis and SCI patients with NDO who completed 52 weeks and were refractory to ≥1 anticholinergic, received Botox 200U or 300U injection or multiple treatments during the extension study. The episodes of urinary incontinence/day reduced by −3.2 to −4.1 across six treatments; the voided volume consistently increased, nearly doubling after treatment and Incontinence Quality of Life (I-QoL) consistently improved more than twice. The results were similar with the 200U and 300U injection. The mean duration of effects was 9 months for patients who received 200U Botox only. Urinary tract infection (UTI) and urinary retention were the most common adverse events. De novo CIC rates in patients with 300U Botox were higher than 200U [48]. Another randomized controlled trial including 72 SCI patients with NDO reported that a 200U or 300U Botox injection had led to a similar improvement in incontinence episodes and QoL. Uninhibited DO improved more in the 300U Botox group at the end point (*p* = 0.01) [49].

A meta-analysis study included six randomized controlled trials involving 437 NDO and idiopathic DO patients, who underwent trigonal-involved or trigonal-sparing Botox injection. The trigone involving group demonstrated a significant improvement in symptom score, complete dryness, incontinence episodes, detrusor pressure at maximum flow rate and volume at the first desire to void compared to the trigonal sparing group. The adverse events in both groups were similar and the trigonal-involving injection did not cause vesicoureteral reflux [50].

## 4. Management of Voiding Dysfunction

### 4.1. Catheterization

CIC remains the mainstay and minimally invasive treatment for neurogenic bladder. CIC for SCI patients with hydronephrosis, recurrent UTI and large PVR remains the gold standard rather than free micturition [51]. The frequency of CIC depends on the fluid intake and patient’s safe bladder capacity. There is no definite number of CIC per day in the current guidelines of NLUTD; however, CIC every 4–6 h is usually recommended for patients with complete urinary retention. Amongst patients with SCI and NLUTD requiring catheter-based drainage, the use of CIC is associated with lower rates of UTI than indwelling urethral catheter (IUC). Comparisons of IUC versus suprapubic catheter (SPC) and SPC vs. CIS gave mixed results [52]. Previous studies showed clean CIC as non-inferior to sterile CIC as it is more cost-effective and convenient for the patient [53]. In selected patients who demand a continent catheterizable cystostomy, constructing a continence channel using appendix, terminal ileum, or a nipple valve is feasible and the patient usually has a high satisfaction rate [54]; however, the long-term complications should be cautiously monitored [55].

An indwelling catheter can also be considered when patients have troublesome motor dexterity, spasticity, or lack of reliable caregiver for CIC support. Suprapubic catheters are considered to prevent urethral trauma and permit sexual function, and have been recommended when CIC is not feasible for patients with chronic SCI without available hand function [56]. Krebs et al. performed a retrospective investigation on the association between bladder management and occurrence of symptomatic UTI in 1104 chronic neurogenic LUTD patients, with a mean duration of 20.3 ± 11.6 years [57]. The study revealed that indwelling catheters caused the highest occurrence of UTI and recurrent UTI, 10- and 4-fold, respectively. The occurrence rate of symptomatic UTI in indwelling catheter was 83.3%, CIC 39%, suprapubic catheter 11%, triggered reflex voiding 27%. Therefore, suprapubic catheterization is more recommended for SCI patients rather than CIC and the least recommended are chronic indwelling catheters.

### 4.2. Triggered Voiding

Triggered reflex voiding can be performed by stimulation of sacral and lumbar dermatomes (such as thigh scratching and suprapubic tapping), resulting in provocation of bladder contractions. Bladder expression by Valsalva or Credé maneuvers (straining/external compression) is not recommended as it is associated with a rise of intravesical pressures. Both management procedures are only recommended in patients with a urodynamically safe bladder [58].

### 4.3. Alpha-Adrenoceptor Blockers

α-adrenoceptor blockers are a mainstay treatment for male benign prostate obstruction. α-blockers are believed to relax the internal urethral sphincter and smooth muscle component of prostate, urethral and bladder neck in men, resulting in reduced bladder outlet obstruction and facilitating voiding in a selected population that maintain the ability to void [17]. In a small group study on 12 SCI patients with high intravesical pressure and poor compliance, terazosin 5 mg daily was administered for 4 weeks. Urodynamic detrusor pressure at maximum capacity decreased by a mean of 36 cm H_2_O (*p* < 0.001) for a 73% improvement in compliance compared to baseline. This effect was reversible with cessation of the medication [59]. α-blockers improve autonomic dysreflexia (AD) symptoms in SCI patients. Terazosin reduces the frequency of AD episodes and the severity of AD symptoms and has no impact on erective function in a prospective trial [60]. A randomized controlled trial reported patients with injuries above T6 had 44% becoming AD symptom free after taking Tamsulosin [61].

### 4.4. Botox Injection into Urethral Sphincter Muscle

Urethral sphincter Botox injection results in the relaxation of the urethral sphincter muscle during the voiding phase. SCI patients with suprasacral lesions frequently presented with NDO and DSD. Shokhl et al. performed a systematic review to examine the efficacy and outcome of urethral sphincter Botox injection. The review included 11 studies and 353 patients. Botox was effective in 60–78% of patients in reducing PVR, mean detrusor pressure, detrusor leak point pressure, and mean urethral pressure 1 month after injection, and most of the patients needed reinjection after an average 4 to 9 months [62]. Our previous study included 33 SCI patients with DSD treated with transurethral sphincter injection of 100U of Botox. The overall satisfaction was 60.6%; however, the frequent occurrence of incontinence became the major dissatisfaction among the patients. Patient selection is crucial for a better outcome [63]. In another previous study, we investigated 118 SCI patients with dysuria [64]. Patients with cervical SCI, DO with DSD, partial hand function and incomplete SCI had a better improvement rate. Only 35.6% of the urge urinary incontinence patients received repeated urethral sphincter Botox injections while more than 60% of patients converted to augmentation enterocytoplasty, bladder outlet surgery, intradetrusor Botox injection or medication, and eventually had moderate to marked voiding improvement. Only certain SCI patients benefitted from urethral sphincter Botox injection, and other bladder management might have had a better clinical outcome.

## 5. Active Surveillance of Urinary Tract Function and Active Treatment in Chronic SCI Patients

Bladder and urethral dysfunction changes with time in patients with NLUTD. Nearly half of the patients with untreated DSD can develop urological complications such as high intravesical pressure, urolithiasis, UTI, vesicoureteral reflux, hydronephrosis, obstructive uropathy and renal failure [65]. These patients should be regularly followed up for lower urinary tract dysfunction and any urological complication should be adequately treated. Although chronic SCI patients with LUTD may be properly diagnosed and treated, all patients should receive life-long surveillance to prevent the development of urological complications and undesired LUTS [66,67,68]. The recommended evaluation procedures vary widely among different centers. In our institution, we recommend: (1) possible UTI testing if turbid urine or hematuria occur, as checked by the patient by dipstick, (2) urinalysis every second month (optional), (3) upper urinary tract, bladder morphology, and PVR check every six months by ultrasound, (4) physical examination, blood chemistry and urine laboratory tests every year, (5) detailed investigation by a specialist every 1–2 years and on demand when risk factors emerge. The investigation is specified according to the patient’s actual risk profile, but should include a video urodynamics investigation for the high-risk patients and be performed in a leading neuro-urological center (optional). (6) More frequent investigations are needed in high-risk patients or if the neurological pathology or the NLUTD status demand this. The need for evaluation items should be based on the risk of patients with chronic SCI and their lower urinary tract conditions [69,70,71,72,73].

Patients with SCI are at an increased risk of bladder cancer and are more likely to be diagnosed at a later stage. The incidence of bladder cancer in SCI patients is 6‰ and the mean onset time after SCI is 18–34 years in earlier reports [74,75]. Indwelling catheters for more than 10 years, and neurogenic bladder itself, are thought to be risk factors for developing bladder cancer, and a highly aggressive tumor subtype, squamous cell bladder cancer, was found in SCI patients. [71,76]. Under this consideration, we recommend regular urology clinic follow-up for every SCI patient, and avoiding long time indwelling catheters if possible.

Avoiding a chronic indwelling catheter can also reduce the incidence of developing a low compliant bladder. Long-term antimuscarinic therapy can decrease urinary incontinence and lower intravesical pressure [58]. Intravesical instillation of vanilloids and BoNT-A injections are alternative treatments for refractory DO or low compliant bladder and can replace the need for bladder augmentation [70]. When surgical intervention is necessary, less invasive types of surgery and reversible procedures should be considered first and any unnecessary surgery in the lower urinary tract should be avoided [52,58,70]. Keeping the bladder and urethra in good condition without interference with neuromuscular continuity provides patients with NLUTD a chance to try new technologies in the future. Improving the quality of life in patients with NLUTD is the most important aspect of treatment. The flow chart of conservative bladder management for patients with chronic SCI and different NLUTD, the goals of bladder management, and the items of active surveillance for bladder and renal function are shown in Figure 1.

There is a need for partnership between urologists, nephrologists and physiatrists to promote and ensure the health of SCI patients.

To achieve successful bladder management and prevention of upper urinary tract deterioration in patients with chronic SCI, teamwork between urologists, rehabilitation doctors, nephrologists, and physiatrists is mandatory. In clinical practice, most SCI patients are handled by rehabilitation doctors and receive physical therapy by physiatrists. In order to have stable bladder and normal upper urinary tract function, a low intravesical pressure and adequate bladder emptying by constant urinary drainage are necessary [77,78]. Regular follow-up of bladder function to ensure the maintenance of a safe functional bladder capacity, prescribing appropriate medications to treat urinary incontinence or bladder outlet dysfunction by urologists, annual check-up of renal function and absence of hydronephrosis by nephrologists, and suitable bladder management methods by CIC without large PVR, performing the Crede maneuver or abdominal tapping, or nerve stimulation therapy by rehabilitation doctors and physiatrists are all important aspects in the long-term care of patients with chronic SCI. When conservative bladder management fails to protect the upper urinary tract, surgical intervention should be performed by urologists to protect the upper urinary tract function.

## 6. Conclusions

NLUTD is common among SCI patients and has a pronounced effect on their health and QoL. The level of neurological lesion affects their pattern of LUT dysfunction. Patients with NLUTD and inappropriate management can have urological complications including UTI, hydronephrosis, and renal failure. Therefore, annual renal sonography workups and regular video urodynamics examinations are crucial to preserve upper urinary tract functions. Despite extensive data on neurogenic bladders, there are relatively few novel publications and there is a lack of high-quality evidence. There is a paucity of new minimally invasive and prolonged efficacy treatments for neurogenic bladder, and partnership between urologists, nephrologists and physiatrists is needed to promote and ensure the health of SCI patients in the future.

## Figures and Tables

**Figure 1 jcm-12-02021-f001:**
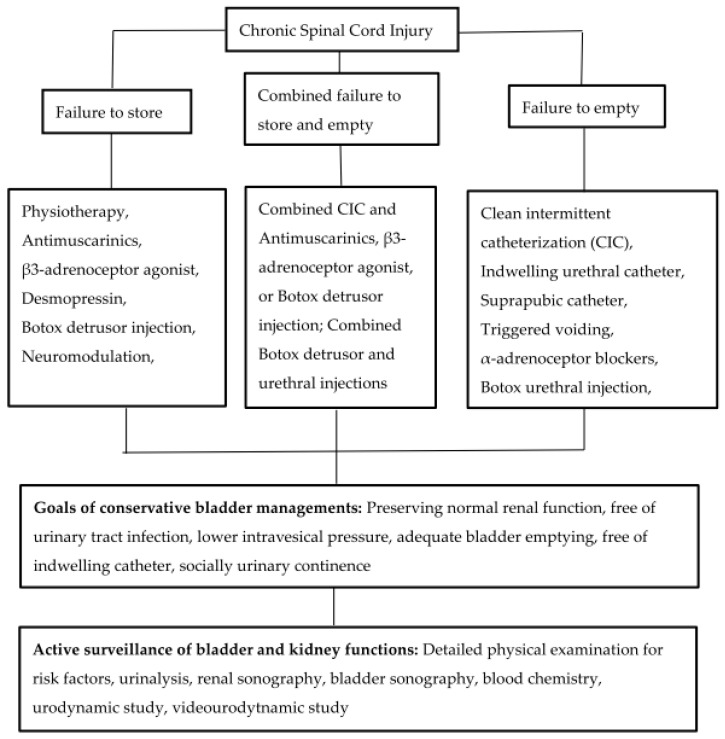
The flow chart of conservative bladder management for patients with chronic SCI and different NLUTD, the goals of bladder management, and the items of active surveillance for bladder and renal function.

## Data Availability

Data are available by contacting with the corresponding author.

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
