# Peer review of "Conservative Bladder Management and Medical Treatment in Chronic Spinal Cord Injury Patients"

_jcm, 2023, doi:10.3390/jcm12052021_

Round 1

Reviewer 1 Report (Previous Reviewer 1)

Thank you for responding to the previous comments. The only issue that needs careful revision is the study method. Deleting the description of the method used to draft the "review" to avoid commenting on the selection process does not sound practical. The authors still included a statement judging the literature in their abstract "Despite the extensive evidence base on NLUTD, there are still relatively few novel publications and lack of high-quality evidence.". Using such a statement without showing a clear methodology to evaluate the evidence level of the reviewed literature reduces the credibility of the paper. Here, I would like to highlight that not only metanalysis needs a protocol for literature selection, all kinds of review (systemic, scoping or systematized..etc) should follow the same scheme to a certain degree. 

Thank you 

Author Response

Reviewer #1:

Thank you for responding to the previous comments. The only issue that needs careful revision is the study method. Deleting the description of the method used to draft the "review" to avoid commenting on the selection process does not sound practical. The authors still included a statement judging the literature in their abstract "Despite the extensive evidence base on NLUTD, there are still relatively few novel publications and lack of high-quality evidence.". Using such a statement without showing a clear methodology to evaluate the evidence level of the reviewed literature reduces the credibility of the paper. Here, I would like to highlight that not only metanalysis needs a protocol for literature selection, all kinds of review (systemic, scoping or systematized..etc) should follow the same scheme to a certain degree.

Reply: Thank you for the comment. We agree with you that the literature search statement is important to declare the source of references used in this review. We have added some statement regarding the scope of literature search. (Lines 76-84)

Reviewer 2 Report (New Reviewer)

Overall very well organized and an interesting read/contribution to the literature. There are some grammatical errors that would be improved with some extra eyes reading the paper for clarity.

In the introduction where you discuss the LUT dysfunction associated with neurological lesions, I might add words like "may" and "generally" to emphasize that the lesion does not always correlate with the expected storage/voiding pattern. Just be a bit less definitive in your wording.

Maybe add a sentence about the benefits of trospium in the elderly.

In your paragraph about neuromodulation, what's the evidence for interstim (sacral neuromodulation) in the SCI population?

In the paragraph about botox, make sure you define what NDO is before using the acronym. Same with CIC (you have IC defined but not CIC)

Add a reference for the recommendation of CIC 1-3x/day or 4-6x/day if there is one. If not, note that it is the recommendation of the authors/at your institution so not to assume that it is a well-studied recommendation.

Although this is clearly a conservative/medical management review, you may want to add a sentence about considerations of a continent catheterizable channel when you discuss SPT in the catheterization section.

In the active surveillance paragraph, I'd be careful about making recommendation statements without a) a reference or b) noting that it is YOUR recommendation. For example, we wouldn't check urinalyses for UTI, but for proteinuria at our institution. We also don't routinely get UDS, only if there is a change in the US or patient symptoms after a period of stability. You can make recommendations, just make sure it's clear that they are yours and not based on a consensus.

Consider adding in a paragraph about cancer surveillance in the setting of chronic catheterization.

Author Response

Reviewer #2

Overall very well organized and an interesting read/contribution to the literature. There are some grammatical errors that would be improved with some extra eyes reading the paper for clarity.

Reply: Thank you for the comment. We have corrected some grammatical errors in the text.

In the introduction where you discuss the LUT dysfunction associated with neurological lesions, I might add words like "may" and "generally" to emphasize that the lesion does not always correlate with the expected storage/voiding pattern. Just be a bit less definitive in your wording.

Reply: Thank you for the comment. We have rephrased the statements in Introduction to emphasize a less definite relationship between neurological lesions and the lower urinary tract conditions. (Lines 55, 59, 60, 62, 63-64, 66-67, 70)

Maybe add a sentence about the benefits of trospium in the elderly.

Reply: Thank you for the comment. We have added a sentence that trospium has a lower risk of cognitive impairment in the elderly. (Lines 120-123)

In your paragraph about neuromodulation, what's the evidence for interstim (sacral neuromodulation) in the SCI population?

Reply: Thank you for the comment. In the last version of manuscript, we did not report the evidence of SNM (interstim) in spinal cord injury. In the revised manuscript, we had added a paragraph to briefly report the clinical efficacy of SNM on NLUTD, and stated that SNM had limited efficacy in complete SCI, but was reported effective if early treatment was given in SCI patients at spinal shock phase. (Lines 183-189)

In the paragraph about botox, make sure you define what NDO is before using the acronym. Same with CIC (you have IC defined but not CIC)

Reply: Thank you for the comment. We have revised the sentence and use non-neurogenic DO for the statement of results from reference 44, (Line 200) The abbreviation of CIC is consistently used in the text. (Line 231)

Add a reference for the recommendation of CIC 1-3x/day or 4-6x/day if there is one. If not, note that it is the recommendation of the authors/at your institution so not to assume that it is a well-studied recommendation.

Reply: Thank you for the comment. We have revised the statement of the daily frequency of CIC. The frequency of CIC depends on the fluid intake and patient’s safety bladder capacity. There is no definite number of CIC per day in the current guidelines of NLUTD, however, CIC every 4-6 hours is usually recommended for patients with complete urinary retention. (Lines 233- 236)

Although this is clearly a conservative/medical management review, you may want to add a sentence about considerations of a continent catheterizable channel when you discuss SPT in the catheterization section.

Reply: Thank you for the comment. We have added a statement of the continent catheterizable channel in managing NLUTD. (Lines 241-244)

In the active surveillance paragraph, I'd be careful about making recommendation statements without a) a reference or b) noting that it is YOUR recommendation. For example, we wouldn't check urinalyses for UTI, but for proteinuria at our institution. We also don't routinely get UDS, only if there is a change in the US or patient symptoms after a period of stability. You can make recommendations, just make sure it's clear that they are yours and not based on a consensus.

Reply: Thank you for the comment. We have revised the statements in the paragraph of surveillance, including: The recommended evaluation procedures varied widely among different centers. In our institution, we recommend: (1) possible UTI if turbid urine or hematuria occur, checked by the patient by dipstick, (2) urinalysis every second month (optional), (Lines 307-310), and a videourodynamic investigation for the high-risk patients. (Lines 314-315)

Consider adding in a paragraph about cancer surveillance in the setting of chronic catheterization.

Reply: Thank you for the comment. We have added statement about the increased risk of bladder cancer in SCI patients with chronic indwelling catheter. Patients with SCI are at increased risk of bladder cancer and are more likely to be diagnosed at a later stage. The incidence of bladder cancer in SCI patients is 6‰ and the mean onset time after SCI is 18-34 years in earlier reports [66, 67]. Indwelling catheters for more than 10 years, and neurogenic bladder itself, are thought to be risk factors for developing bladder cancer, and a highly aggressive tumor subtype, squamous cell bladder cancer, was found in SCI patients. [68,69]. Under this consideration, we recommend regular urology clinic follow-up for every SCI patient, and avoiding long time indwelling catheter if possible. (Lines 320-327)

This manuscript is a resubmission of an earlier submission. The following is a list of the peer review reports and author responses from that submission.

Round 1

Reviewer 1 Report

Dear Authors,

I have read the article and you have selected an important topic in the field of urology. There are certain points that need your kind attention to improve the readability of the article:

- English revision is needed as there are several grammatical issues that make it sometimes confusing to read.

- The abstract and the conclusion included a statement about the evaluation of the SCI patients with NLUTD, but this aspect was not covered in the manuscript. The authors might elaborate on the recommended evaluation/follow-up for their patients according to the best practices, which matches the way they are discussing the NLUTD.

- The authors did not state clearly the type of this article, though that is it clearly a review, it need to be highlighted.

- Given the previous point, the review methodology is not specific and several aspects are missing in that regard: number of articles, inclusion and exclusion criteria, strength of evidence that was accepted (especially that the authors commented twice on the quality of the published papers, so their evaluation should be authentic to avoid misjudgment).

- The introduction: Despite including a detailed description on the subject matter, ONLY 2 references were cited for the whole introduction!

- The aim was to underpin or review the recent evidence on the non-surgical treatment of NLUTD by SCI, but about 10 references were beyond the search range given 2002-2022 (here I believe the authors meant 2022, as it is written 20202). There is no harm here, but a note on the use of these relatively old references need to be there. 

- The review lacks a strong recommendation or a clear evaluation of the evidence for the use of each method (some lines of treatment, such as the pharmacological treatment of the storage dysfunction, are very brief). There are some recent reviews that the authors can refer to to enhance the discussion and provide up-to-date evaluation of the matter (one example is here: https://doi.org/10.1007/s00345-018-2419-z, which discusses the very same theme of the current article).

-  The statement about the partnership between the three teams for the treatment: would the authors write a paragraph at the end of the manuscript (before the conclusion) to explain the reason for such a recommendation?

- A graphical presentation of the discussed methods/approaches would improve the readability of the manuscript. However, it should be supported with evidences.

Thank you 

https://doi.org/10.1007/s00345-018-2419-z

Reviewer 2 Report

This is an updated review about non-surgical management for neurogenic lower urinary tract dysfunction (NLUTD) in patients with chronic spinal cord injury (SCI). Lower urinary tract dysfunction might lead to urological complications such as high intravesical pressure, urolithiasis, urinary tract infections, vesicoureteral reflux, hydronephrosis, obstructive uropathy and renal failure. However, the authors showed that there are still relatively few novel publications and there is lack of high-quality evidence about non-surgical therapies of NLUTD and for this reason new minimally invasive and effective treatments are needed. In this context, a close partnership between urologists, nephrologists and physiatrists is required. Overall, this review provides an up-to-date picture of the state of the art of conservative and medical treatment for chronic SCI patients. I think that only some little corrections should be made by the authors. For example, in line 77, the sentence “We searched PUBMED articles published from January 2002 to August, 20202” should be corrected as follows: “We searched PUBMED articles published from January 2002 to August, 2022”. Moreover, in line 273 there is a typo, in fact it is written "pronounce effect" instead of “pronounced effect”. Globally, I think that a spelling and punctuation check should be performed.

Author Response

  1. I think that only some little corrections should be made by the authors. For example, in line 77, the sentence “We searched PUBMED articles published from January 2002 to August, 20202” should be corrected as follows: “We searched PUBMED articles published from January 2002 to August, 2022”.

Ans.: I have edited the eror.

2.  Moreover, in line 273 there is a typo, in fact it is written "pronounce effect" instead of “pronounced effect”. Globally, I think that a spelling and punctuation check should be performed.

Ans.: I have edited the eror and English revision was made.

Thank you so much :)